sRESEARCH ARTICLE

# Effectiveness of utilizing the WHO safe childbirth checklist on improving essential childbirth practices and maternal and perinatal outcome: A systematic review and meta-analysis

**Lemi Belay Tolu** [1]*, **Wondimu Gudu Jeldu** [1], **Garumma Tolu Feyissa** [2]

1 Department of Obstetrics and Gynaecology, Saint Paul's Hospital Millennium Medical College, Addis Ababa, Ethiopia, 2 Department of Health, Behaviour, and Society, Jimma University, Jimma, Ethiopia

* lemi.belay@gmail.com

**Data Availability Statement:** All relevant data are within the paper and its Supporting Information files.

## Abstract

### Introduction

The World Health Organization (WHO) Safe Childbirth Checklist (SCC) is a 29-item checklist based on essential childbirth practices to help health-care workers to deliver consistently high quality maternal and perinatal care. The Checklist was intended to reduce maternal and perinatal mortality and address the primary cause of maternal death, intrapartum stillbirth, and early neonatal death. The objective of this review was to locate international literature reporting on the effectiveness of utilizing the WHO safe childbirth checklist on improving essential childbirth practices, early neonatal death, stillbirth, maternal mortality, and morbidity.

### Methods

We searched MEDLINE, google scholar, Cochrane Central Register of Controlled Trials (CENTRAL), met-Register of Controlled Trials (*m*-RCT) (www.controlled-trials.com), ClinicalTrials.gov (www.clinicaltrials.gov) and the WHO International Clinical Trials Registry Platform (ICTRP) (www.who.int/stop/search/en) to retrieve all available comparative studieshttp://www.opengrey.eu/ published in English after 2008. Two reviewers did study selection, critical appraisal, and data extraction independently. We did a random or fixed-effect meta-analysis to pool studies together and effect estimates were expressed as an odds ratio. Quality of evidence for major outcomes was assessed using the Grading of Recommendations, Assessment, development, and evaluation(GRADE).

### Results

We retained three cluster randomized trials and six pre-and-post intervention studies reporting on WHO SCC's. The WHO SCC utilization improved quality of preeclampsia management(moderate quality of evidence) (OR = 7.05 [95% CI 2.34–21.29]), maternal infection management(moderate quality of evidence) (OR = 7.29[95%CI 2.29–23.27]), Partograph

**Funding:** The author(s) received no specific funding for this work.

**Competing interests:** The authors have declared that no competing interests exist.

utilization(moderate quality of evidence) (OR = 3.81 [95% 1.72–8.43]), postpartum counselling(low quality of evidence) (RR = 132.51[95% 49.27–356.36]) and still birth(moderate quality of evidence) (OR = 0.92[95% CI 0.87–0.96]). However, the utilization of the checklist had no impact on early neonatal death (very low quality of evidence) (OR = 1.07[95%CI [1.01–1.13]) and maternal death (low quality of evidence) (OR = 1.06[95% CI 0.77–1.45]).

## Conclusions

Moderate quality of evidence indicates that WHO SCC utilization is effective in reducing stillbirth and Improving preeclampsia management, maternal infection management and partograph utilization Low quality of evidence indicates that WHO SCC is effective in enhancing postpartum danger sign counseling. Low and very low quality of evidence suggests that WHO SCC has no impact on maternal and early neonatal death, respectively.

## Introduction

From more than 130 million births per year, the World Health Organization (WHO) estimates nearly 2,87,000 maternal deaths, 1 million intrapartum related stillbirths, and 3 million newborn deaths during the neonatal period [1]. The majority of maternal and perinatal deaths are clustered around the time of birth, with the highest number of deaths occurring within the first 24 hours after childbirth [1]. As a solution to reduce this high perinatal mortality the World Health Organization (WHO) has introduced a safe childbirth checklist(SCC) in 2008, a 29-item evidence-based essential childbirth practice to help health-care workers to deliver consistently high quality maternal and perinatal care [2]. The WHO Safe Childbirth Checklist (SCC) incorporates major causes of maternal death, intrapartum stillbirth, and early neonatal death and expected to have an impact on maternal and perinatal morbidity and mortality [3]. These practices are organized on admission, just before birth, soon after birth, and on discharge to confirm health care workers have completed essential birth practices at each point for every birth event [4].

A pre and post-intervention pilot studies in India reported a two-fold increased delivery of evidence-based essential birth practices after the introduction of SCC compared to practices before the introduction of the SCC at each birth event [5]. However, some studies report that SCC has no impact on perinatal or maternal mortality [3]. Another study in a tertiary center in Sri Lanka stated that the SCC tool uptake by healthcare workers was higher and has resulted in improved delivery of evidence-based birth practices [6].

Observational studies reported significant improvement in WHO SCC targeted essential maternal and newborn care practices [7–11]. The studies indicate that women delivering in WHO SCC program intervention facilities received more safe childbirth practices as compared to women receiving care in the control facilities [7–11].

A prospective interventional study conducted at a tertiary care hospital in India found that implementation of a safe childbirth checklist has no impact on maternal or neonatal mortality reduction. However, there was increased partograph use, antibiotic administration, and active management of the third stage of labour [12]. The Better-Birth trial in north India, where peer coaching was used to increase adherence of workers to WHO SCC at sub-district and primary health care facilities, reported a significant increase in health care worker's adherence to essential practices. However, the study indicated that the utilization of the tool didn't reduce

perinatal and maternal death [13]. A recent quasi-experimental study conducted in the Rajasthan district of India found out that implementation of SCC program potential averts 40,000 intrapartum deaths per year, the most reduction being from prevention of stillbirths [14].

Contradicting results from different studies on WHO SCC's impact on maternal and perinatal death despite an improvement of essential practices mandates searching for robust evidence on the effectiveness of SCC implementation on improving essential childbirth practices and reduction of maternal and perinatal deaths. In our review, Therefore, this systematic review aimed to investigate the effectiveness of utilizing the WHO safe childbirth checklist on improving essential childbirth practices and maternal and perinatal outcomes.

### Review question(s)

The review sought to locate international literature reporting on the impact of WHO SCC utilization. Specifically, the review questions were:

- What is the effectiveness of the WHO safe childbirth checklist on improving essential childbirth practices?

- What are the effectiveness of the WHO safe childbirth checklist on reducing maternal and perinatal morbidity and mortality?

## Methods

This systematic review was prepared using PRISMA reporting guidelines (S1 Checklist) for systematic reviews [15]. The review was conducted per Cochrane handbook for a systematic review of interventions [16], and a prior protocol registered in PROSPERO 2019 CRD42019137092(available at https://www.crd.york.ac.uk/PROSPERO/display_record.php?RecordID=137092). During the conduct of the review, we considered the following inclusion criteria:

### Participants

For the sake of this review, we considered health professionals directly involved in the care for mothers and newborns during labour, delivery, and post-partum periods and mothers and newborns in any health care settings.

### Intervention

The intervention we considered for this review was the utilization of the WHO safe childbirth checklist by health professionals.

### Comparator

The comparator considered for this review was labouring mothers and newborn care without WHO safe childbirth or any other structured checklist.

### Outcomes

The outcomes considered for this review were the incidence of essential childbirth practices, early neonatal death, stillbirth, maternal death, and maternal morbidity.

Essential childbirth practices considered in this review were:

1. Partograph use.

2. Maternal infection management upon admission: Evaluation of mothers for the necessity of antibiotics by temperature measurement and looking for the sign of intra-amniotic infection like prolonged foul-smelling vaginal discharge, uterine tenderness, and maternal tachycardia.

3. Preeclampsia management: Evaluation of mothers for the necessity of Mgso4 and anti-hypertensive administration by measuring blood pressure upon admission.

4. Active management the third stage of labour (AMTSL): Oxytocin administration within one minute of delivery of baby, controlled cord traction, and uterine massage.

5. Maternal postpartum bleeding assessment.

6. Breastfeeding started within one hour.

7. Newborn feeding assessment upon discharge.

8. Postpartum danger signs counseling.

9. Counseling on family planning.

**Early Neonatal Death (END).**   Death of newborn within seven days of delivery.

**Stillbirth.**   Intrapartum fetal death after the admission of the patient for labour and delivery. Studies that included fetal death before admission of the patient to a health facility were excluded. For this review, we defined perinatal mortality as intrapartum stillbirth and newborn death within seven days of delivery

**Maternal death.**   the death of mothers caused by obstetric related events within the health facilities.

**Maternal morbidity.**   blood transfusion, hysterectomy, maternal sepsis, postpartum bleeding, and maternal seizure.

## Types of studies

This review considered all studies with comparative designs, such as randomized controlled trials (RCTs), and, before and after studies published from 2008 to November 11/2019(the day literature search was done) in English. This date range was selected because the WHO safe childbirth checklist was introduced in 2008 [2].

## Search strategy

We did a preliminary search of PROSPERO, MEDLINE, the Cochrane Database of Systematic Reviews, and the JBI Database of Systematic Reviews and Implementation Reports, and no published or ongoing systematic reviews on the topic were identified. The search strategy aimed to locate both published and unpublished studies. An initial limited search of MEDLINE was undertaken, followed by an examination of the text words contained in the titles and abstracts of relevant articles, and the index terms used to describe the articles. A second search using all identified keywords and index terms was then undertaken across all included databases. Thirdly, the reference list of all identified reports and articles was searched for additional studies. The data basis searched were: MEDLINE, Cochrane Central Register of Controlled Trials (CENTRAL), met-Register of Controlled Trials (*m*-RCT) (www.controlled-trials.com), ClinicalTrials.gov (www.clinicaltrials.gov) and the WHO International Clinical Trials Registry Platform (ICTRP) (www.who.int/ictrp/search/en). Likewise, a search for grey literature was conducted using Google Scholar, Open-Grey (System for Information on Grey

Literature in Europe) (www.opengrey.eu/), and WHO websites. A detailed search strategy for MEDLINE was provided in a supplementary file (S1 Table).

## Study selection

Following the search, all identified citations were loaded into EndNote, and duplicates were removed. Two independent reviewers screened titles and abstracts for assessment against the inclusion criteria for the review. The full texts of potentially eligible studies were retrieved and assessed in detail against the inclusion criteria by two independent reviewers.

## Assessment of methodological quality

Eligible studies were critically appraised by two independent reviewers for methodological quality, using Cochrane risk of bias assessment tool from Rev man [16]. All disagreements that arose were resolved through discussion and, there was no requirement for a third reviewer. All studies regardless of the results of their methodological quality were undergone data extraction, and the results of critical appraisal were reported in narrative form and a table.

## Data extraction and synthesis

We extracted data using the Rev Man version 5.3. The relevant information such as population characteristics, authors, study setting, study design, publication year, interventions, and summary of the findings was extracted. Where necessary, we asked primary authors to provide additional information on the articles. Studies were pooled in a statistical meta-analysis using Rev Man version 5.3. Effect sizes were expressed as odds ratios (for dichotomous data), and their 95% confidence intervals were calculated for analysis. We assessed heterogeneity statistically using the $Tau^2$ and $I^2$ tests. We considered $I^2$ tests above 50% as indicative of significant heterogeneity. Besides, the statistical heterogeneity among studies was checked in terms of study settings, sample size, and study design. We conducted leave out analyses by excluding studies with very large or very low effect estimates and different study designs. Also, we compared the random and fixed-effects model, and the decision was made based on the best-fitting model to the data [17].

The certainty of the quality of evidence was assessed using a software package (Grade pro) developed by the Grading of Recommendations, Assessment, Development, and Evaluation (GRADE) [18], group for the following outcomes: Partograph use, preeclampsia management, maternal infection management, postpartum counseling, neonatal death, stillbirth, and maternal death.

## Results

The search yielded a total of 458 records. After removing duplicates, 130 documents were retained for further examination. After screening the titles and abstracts, 13 papers were retained for full-text review. Based on pre-defined inclusion criteria, nine records were included in the systemic review (Fig 1).

From four studies excluded by reason, two (Delaney et al. [19] and Kara et al. [5]) reported on the impact of peer coaching on adherence to WHO SCC and one study (Patabendige, M and Senanayake, H. [20]) reported effects of Sri-Lanka context-specific modified WHO Safe Childbirth Checklist on adherence to WHO SCC. One cross-sectional study was excluded because of the non-comparative nature of the study (Patabendige, M and Senanayake, H. [8]) (S1 Document).

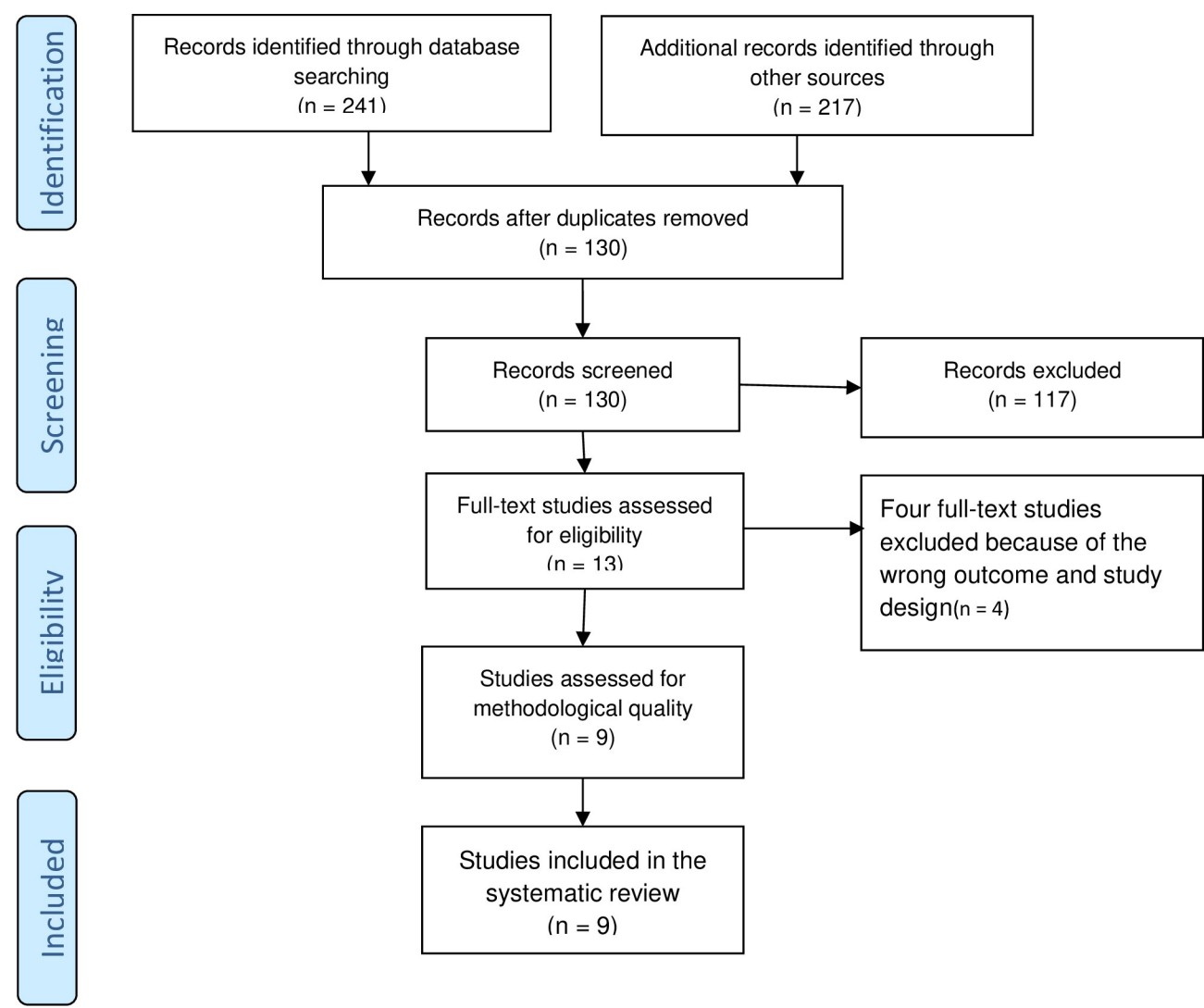

**Fig 1. The prisma flow diagram showing the study selection process.**

## Characteristics of included studies

All the nine studies included compared WHO SCC use to none use of WHO SCC. Among the nine studies included in this review, Varghese et al. [14], Semrau et al. [21], and Kumar et al. [7] reported on the finding of a randomized cluster trial conducted in India. Also, another two pre-and-post intervention studies were conducted in India (Spector et al. [3], and Varaganti et al. [12]). The other four pre-and-post intervention studies were conducted in Italy (Albolino et al. [22]), Namibia (Kabongo et al. [9]), Bangladesh (Nababan et al. [11]), and Rwanda (Tuyishim et al. [10]) respectively. Two studies were conducted at a tertiary health facility (Albolino et al. [22] and Varaganti et al. [12]) whereas four (Kabongo et al. [9], Nababan et al. [11], Tuyishim et al. [10], and Varghese et al. [14]) at the district health facility, two (Semrau et al. [21] and Spector et al. [3]) at the subdistrict health facility and one (Kumara et al. [7]) at both district and subdistrict health facility (Table 1).

**Table 1. Characteristics of included studies.**

| Study ID. | Study design | Setting/country. | Participants | Number of participants in Intervention (WHO SCC)/comparison (Without WHO SCC) groups | Outcomes |
|---|---|---|---|---|---|
| Albolino et al. 2018[22] | pre-and-post intervention. | Tertiary hospital/Italy. | Labouring mothers. | Intervention: 98 | Essential childbirth practices. |
| | | | | Comparasion:141. | |
| Kabongo et al. 2018[9] | pre-and-post intervention | District hospital /Namibia | Labouring mothers and newborns. | Intervention: 1526 | Essential childbirth practices. |
| | | | | Comparasion:1401 | Perinatal outcome. |
| Kumar et al.2016[7]. | Cluster randomized. | District and subdistrict hospital and health centers/ India. | Labouring mothers. | Intervention: 240 | Essential childbirth practices. |
| | | | | Comparasion:240. | |
| Nababan et al., 2017 [11]. | pre-and-post intervention | District Hospital/ Bangladesh. | Uncomplicated vaginal deliveries. | Intervention: 157 | Essential childbirth practices. |
| | | | | Comparasion:153. | |
| Semrau et al., 2017[21]. | Cluster randomized | Subdistrict hospital and primary and community health centers/India. | Labouring mothers and newborns. | Intervention: 1048 | Essential childbirth practices, perinatal outcome, maternal death, and morbidity. |
| | | | | Comparasion:1090 | |
| Spector et al. 2012[3]. | pre-and-post intervention | Subdistrict hospital/India. | Labouring mothers and newborns. | Intervention: 639 | Essential childbirth practices. |
| | | | | Comparasion:405 | Perinatal outcome. |
| | | | | | Maternal death. |
| Tuyishim et al. 2018[10] | pre-and-post intervention | District hospital/Rwanda | Labouring mothers. | Intervention: 95 | Essential childbirth practices. |
| | | | | Comparasion:106 | |
| Varaganti et al. 2018[12]. | pre-and-post intervention. | Tertiary hospital/India. | Labouring mothers and newborns. | Intervention: 620 | Essential childbirth practices. |
| | | | | Comparasion:635 | Maternal death. |
| | | | | | Perinatal outcome. |
| Varghese et al., 2019 [14]. | Cluster-randomized. | District/secondary level facility/India. | Labouring mothers and newborns. | Intervention: 77231 | -Stillbirth. |
| | | | | Comparasion:59800 | -Early neonatal death. |

## The methodological quality of the included studies

Eight and one of the Included studies were judged to be high and low risk for allocation concealment respectively, whereas seven and two of included studies were judged to be high and low risk for random allocation respectively (Fig 2).

**Allocation.** Three studies were cluster-randomized (Varghese et al. [14], Semrau et al. [21], and Kumar et al. [7]). Six studies were-random pre-and-post intervention study (Albolino et al. [22], Kabongo et al. [9], Nababan et al. [11], Spector et al. [3], Varaganti et al. [12] and Tuyishime et al. [10]). Only the study by Kumar et al. (2016) was concealed by central allocation (Fig 3).

**Incomplete outcome data (Attrition Bias).** Only the study by Nababan et al. [11] was felt to have incomplete outcome data potentially, but they did sensitivity analysis by excluding incomplete reports (Fig 3).

## Blinding of participants (performance Bias)

Blinding of health professionals is not possible in all studies as it involves training and introduction of the checklist. Still, three of the studies are cluster randomized with similar data collection for both control and intervention facilities (Varghese et al. [14], Semrau et al. [21] and Kumar et al. [7]) and one study collected retrospectively from documents (Albolino et al. [22]). Five studies collected data by observation of health workers practice which might have introduced hawthorn effect (Albolino et al. [22], Kabongo et al. [9], Nababan et al. [11], Spector et al. [3], Varaganti et al. [12] and Tuyishime et al. [10]) (Fig 3).

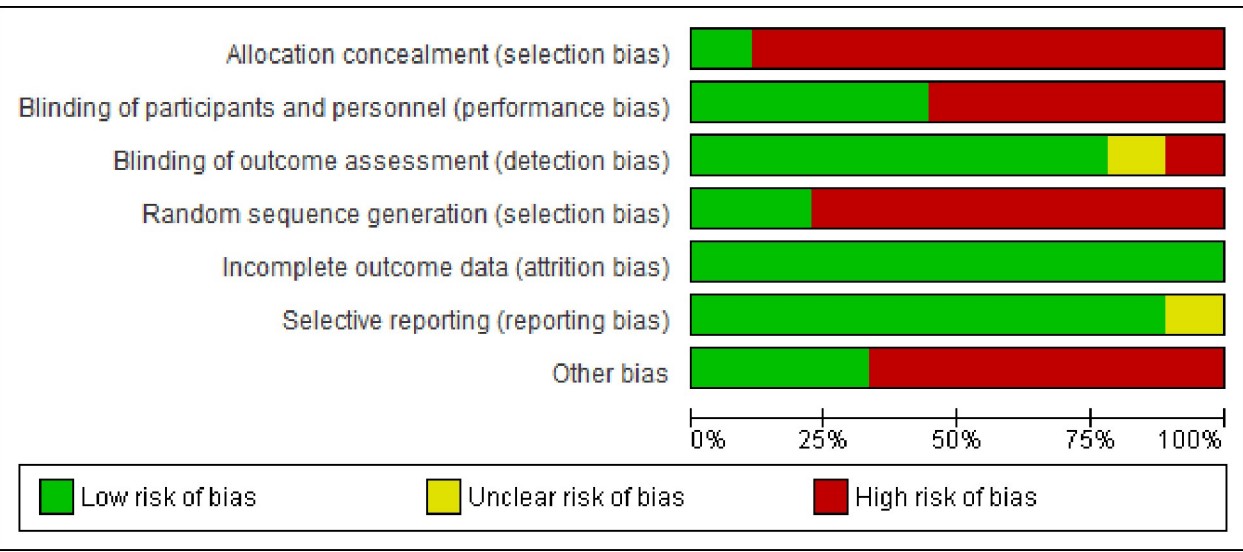

**Fig 2. Risk of bias graph, review authors' judgments about each risk of bias item presented as percentages across all included studies.**

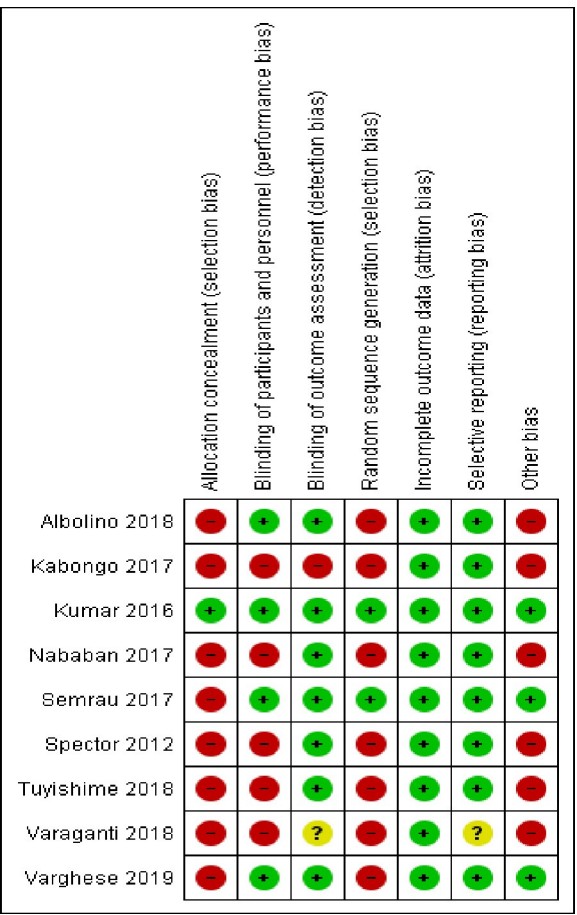

**Fig 3. Risk of bias summary: Review authors' judgments about each risk of bias item for each included study.**

**Blinding of outcome assessment.** Data collectors didn't know intervention and control facilities in three studies (Varghese et al. [14], Semrua et al. [21], and Kumar et al. [7]). Four studies were not blinded but used a pre-defined checklist and unlikely to affect the outcome of the study (Albolino et al. [22], Kabongo et al. [9], Nababan et al. [11], Spector et al. [3], and Tuyishime et al. [10]). In one study, investigators were involved in data collection (Kabongo et al. [9]), and one study didn't report data collection methods (Varaganti et al. [12]) (Fig 3).

**Selective reporting (reporting bias).** Eight studies used a pre-defined data collection protocol, and all outcomes of interest were reported. One study didn't use a clear study protocol (Varaganti et al. [12]) (Fig 3).

**Other potential sources of bias.** Three cluster-randomized studies considered design effect during a sample size calculation, had a control group, and had similar baseline similarity in terms of health professionals (Varghese et al. [14], Semrua et al. [21] and Kumar et al. [7]). Five of the studies were pre-and-post-intervention without a control group and didn't consider the design effect. Still, all had similar baseline health professionals (Albolino et al. [22], Kabongo et al. [9], Nababan et al. [11], Spector et al. [3], Varaganti et al. [12] and Tuyishime et al. [10]) (Fig 3).

## Review findings

**1. Preeclampsia management.** There were seven times more likelihood of evaluating labouring mothers for preeclampsia and administration of MgSo4 and antihypertensive drugs (OR,7.05, 95% CI 2.34–21.29, seven studies, 5667 participants) among professionals utilizing WHO SCC (Moderate quality of evidence). Random effect meta-analysis was used for this outcome because of significant heterogeneity ($I^2 = 97\%$ and $Tau^2 = 1.97$). The substantial heterogeneity indicates that treatment effects vary between studies, so we investigated the factors affecting treatment effects by a subgroup analysis of study design. A meta-analysis of cluster-randomized studies showed 20 times more likelihood of evaluation and management of labouring mothers for preeclampsia among professionals utilizing WHO SCC (OR,20, CI 15.19–26.32, $I^2$ 18% and $Tau^2$ 0.01, two studies, 2618 participants (Fig 4).

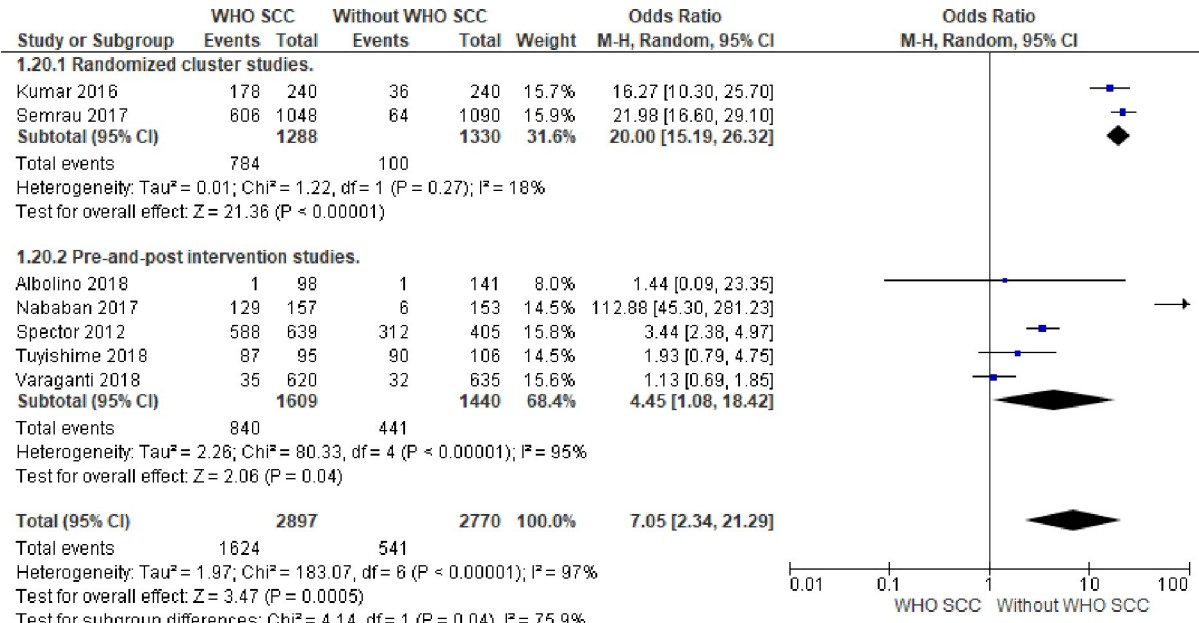

**Fig 4. Forest plot of comparison: 1 WHO SCC use and None use, outcome: 1.1 Preeclampsia management.**

**2. Maternal infection management.**   Professionals utilizing WHO SCC were 17 times more likely to evaluate and manage maternal infection compared to none use of WHO SCC (OR,17.46, 95% CI 3.62–84.24, seven studies, 5667 participants) (Moderate quality of evidence). Random effect meta-analysis was utilized for this outcome because of significant heterogeneity ($I^2$ = 98% and $Tau^2$ = 4.27). The substantial heterogeneity indicates that treatment effects vary between studies, so we investigated the factors affecting treatment effects by a subgroup analysis of study design. A meta-analysis of cluster-randomized studies showed 214 times more likely maternal infection evaluation and management among professionals utilizing WHO SCC (OR 214.85, CI 93.36–494.41, $I^2$ 33% and $Tau^2$ 0.02, two studies, 2618 participants) (Fig 5).

**3. Partograph.**   Professionals utilizing WHO SCC was five times more likely to use partograph compared to none use of WHO SCC (OR 5.48, 95% CI 2.21–13.62, six studies, 5323 participants) (Moderate quality of evidence). Random effect meta-analysis was utilized for this outcome because of significant heterogeneity ($I^2$ = 95% and $Tau^2$ = 1.10) The substantial heterogeneity indicates that treatment effects vary between studies, so we investigated the factors affecting treatment effects by a subgroup analysis of study design. A meta-analysis of cluster-randomized studies showed twelve times more partograph use among professionals utilizing WHO SCC (OR 12.25, CI 7.29–20.59, $I^2$ 0%, two studies, 2618 participants) (Fig 6).

**4. Active Management of the Third Stage of Labour (AMTSL).**   Six studies reported on AMTSL (Kumar et al. [7], Nababan et al. [11], Spector et al. [3], Tuyishim et al. [10], Varaganti et al. [12] and Semrau et al. [21]). We didn't perform a meta-analysis because of significant heterogeneity that did not resolve by planned sensitivity analysis. Four studies uniformly reported significant improvement in AMTSL with WHO SCC utilization (Kumar et al. [7], Spector et al. [3], Varaganti et al. [12], and Semrau et al. [21]). In contrast, two studies reported statistically non-significant improvement (Nababan et al. [11] and Tuyishim et al. [10]) (Table 2).

**5. Maternal postpartum bleeding assessment.**   Four studies reported on maternal postpartum bleeding assessment (Kumar et al. [7], Nababan et al. [11], Spector et al. [3], and

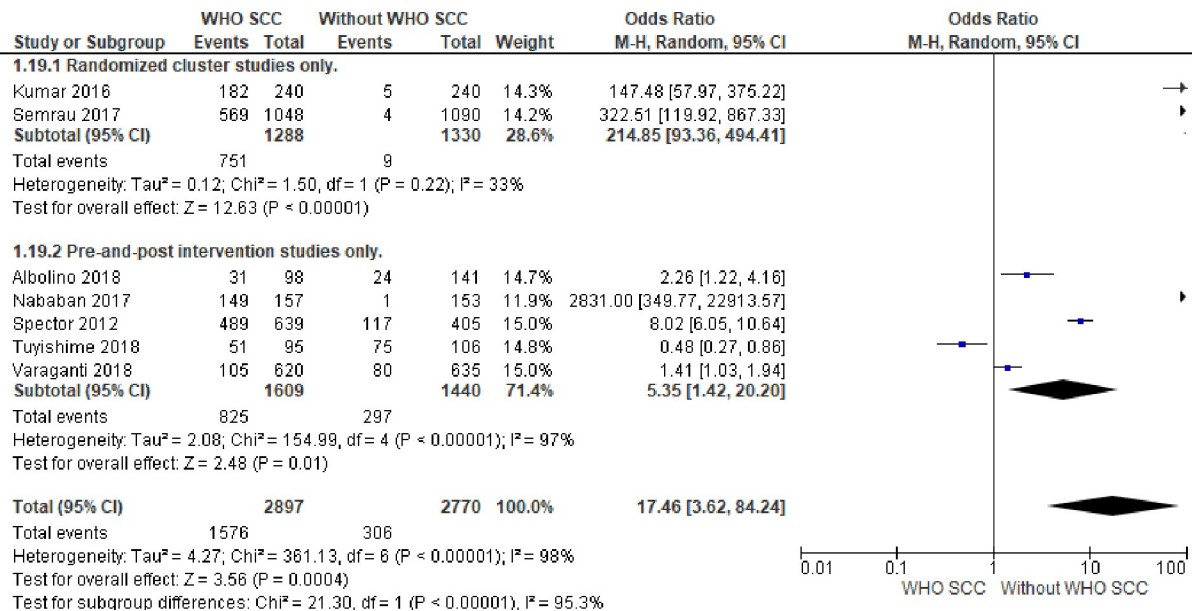

**Fig 5.  Forest plot of comparison: 1 WHO SCC use and none use., outcome: 1.2 Maternal infection management.**

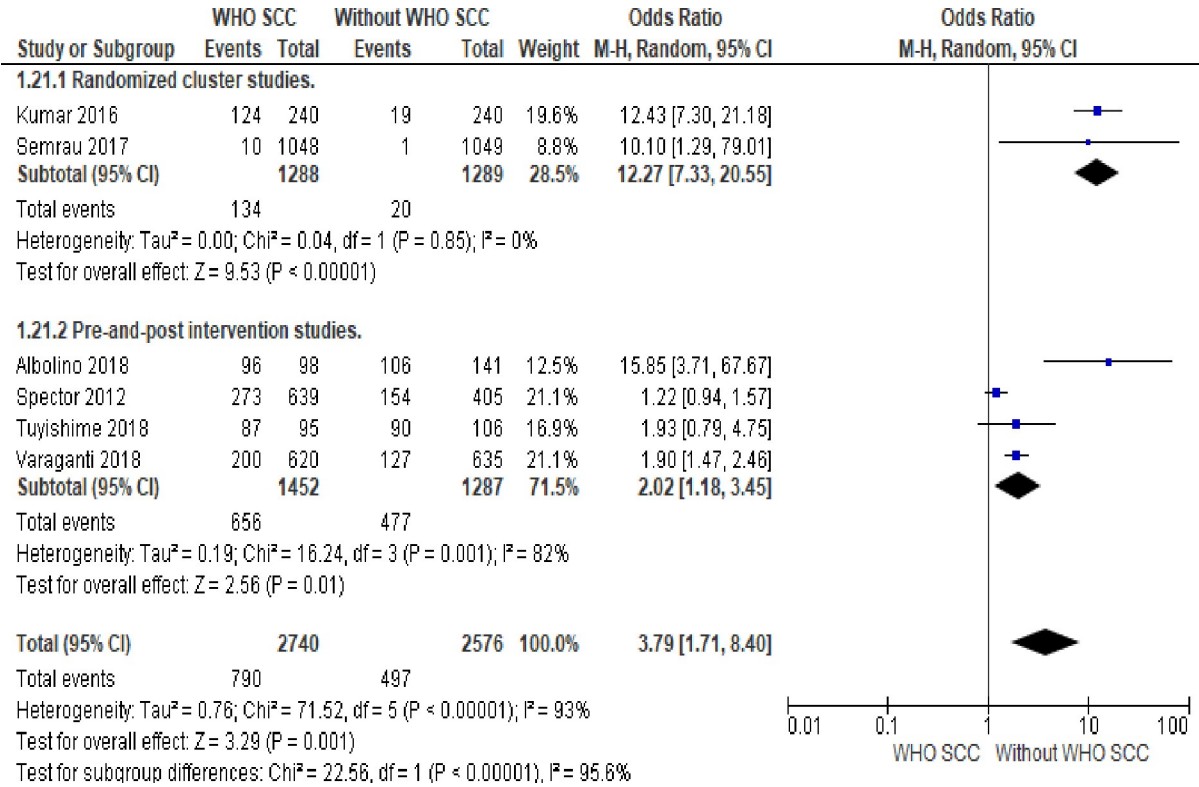

**Fig 6. Forest plot of comparison: 1 WHO SCC use and none use., outcome: 1.3 Partograph.**

Varaganti et al. [12]). We didn't perform a meta-analysis because of significant heterogeneity that did not resolve by planned sensitivity analysis. Two studies uniformly reported significant improvement in postpartum bleeding assessment with WHO SCC utilization (Kumar et al. [7] and Spector et al. [3]). In contrast, two studies reported statistically non-significant improvement (Varaganti et al. [12] and Nababan et al. [11]) (Table 3).

**6. Breastfeeding started within one hour.** Mothers handled by professionals utilizing WHO SCC was 17 times more likely to initiate breastfeeding within one hour compared to none use of WHO SCC (OR 21.18, 95% CI 17.54–25.57, five studies, 4050 participants). Random effect meta-analysis was utilized for this outcome because of significant heterogeneity ($I^2$

**Table 2. AMTSL with and without WHO SCC utilization.**

| Study | AMTSL without WHO SCC (Number / total and %) | AMTSL with WHO SCC (Number/ total and %) | P-value |
|---|---|---|---|
| 1. Kumar et al. [7] | 58/240(24%) | 211/240(88%) | <0.001 |
| 2. Nababan et al. [11] | 134/153(88%) | 156/157(99%) | <0.001 |
| 3. Semrau et al. [21] | 154/1041(14.8%) | 549/1019(53.9%) | <0.001 |
| 4. Spector et al. [3] | 33/338(8.4%) | 402/583(68.9%) | <0.001 |
| 5. Tuyishim et al. [10] | 76/92(77.6%) | 84/98(84.8%) | 0.206 |
| 6. Varaganti et al. [12] | 600/635(94.5%) | 601/620(96.9%) | 0.032 |

**Table 3. Maternal postpartum bleeding assessment with or without WHO SCC utilization.**

| Study | Maternal postpartum bleeding assessment without WHO SCC (Number and %) | Maternal postpartum bleeding assessment with WHO SCC (Number and %) | P-value |
|---|---|---|---|
| 1. Kumar et al. [7] | 84/240(35%) | 218/240(91%) | <0.001 |
| 2. Nababan et al. [11] | 152/153(99%) | 157/157(100%) | 0.318 |
| 3. Spector et al. [3] | 58/388(15%) | 577/583(99%) | <0.001 |
| 4. Varaganti et al. [12] | 25/635(3.94%) | 20/620(3.22%) | 0.498 |

= 89%). The substantial heterogeneity indicates that treatment effects vary between studies, so we investigated the factors affecting treatment effects by subgroup analysis using study design as a grouping variable. A meta-analysis of pre-and-post-intervention studies showed twenty times more likely initiation of breastfeeding among mothers handled by professionals utilizing WHO SCC (OR 20.03, 95%CI 13.38–29.97, $I^2$ 48%, three studies, 1462 participants) (Fig 7).

**7. Newborn assessment for sepsis.** Five studies reported on newborn assessment for sepsis (Kumar et al. [7], Nababan et al. [11], Spector et al. [3], Tuyishim et al. [10] and Varaganti et al. [12]). We didn't perform a meta-analysis because of significant heterogeneity that did not resolve by planned sensitivity analysis. Two studies uniformly reported significant

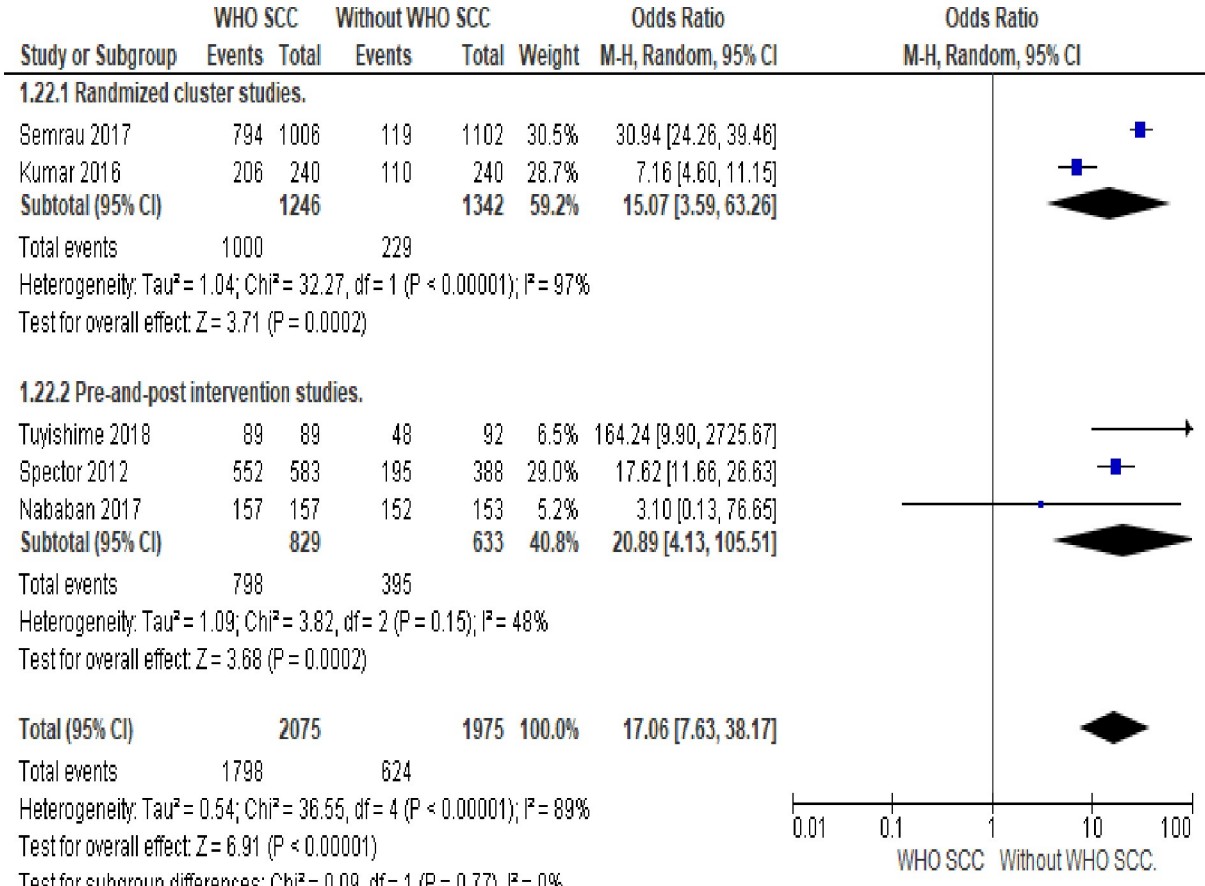

**Fig 7. Forest plot of comparison: 1 WHO SCC use and none use., outcome: 1.4 breastfeeding started within one hour.**

**Table 4. Newborn assessment for sepsis with or without WHO SCC.**

| Study | Newborn assessment for sepsis without WHO SCC (Number and %) | Newborn evaluation for sepsis with WHO SCC (Number and %) | P-value |
|---|---|---|---|
| 1. Kumar et al. [7] | 2/240(1%) | 103/240(43%) | <0.001 |
| 2. Nababan et al. [11] | 1/153(0.65%) | 1/157(0.64%) | 0.985 |
| 3. Spector et al. [3] | 0/338 | 279/489(57.1%) | <0.001 |
| 4. Tuyishim et al. [10] | 1/101(1%) | 5/98(5.1%) | 0.115 |
| 5. Varaganti et al. [12] | 100/635(15.7%) | 98/620(15.8%) | 0.977 |

improvement in newborn assessment for sepsis with WHO SCC utilization (Kumar et al. [7] and Spector et al. [3]). In contrast, two studies reported statistically non-significant improvement (Varaganti et al. [12] and Tuyishim et al. [10]). Nababan et al. [11] reported that there is no difference in newborn assessment for sepsis with or without WHO SCC utilization (0.65% versus 0.64% with a p-value of 0.985) (Table 4).

**8. Newborn feeding assessment upon discharge.** Six studies reported on newborn feeding assessment upon discharge (Kumar et al. [7], Nababan et al. [11], Spector et al. [3], Tuyishim et al. [10], Varaganti et al. [12] and Semrau et al. [21]). We didn't perform a meta-analysis because of significant heterogeneity that did not resolve by planned sensitivity analysis. Three studies uniformly reported significant improvement in newborn feeding assessment upon discharge with WHO SCC utilization (Kumar et al. [7], Spector et al. [3], and Semrau et al. [21]). In contrast, three studies reported statistically non-significant improvement in newborn feeding assessment upon discharge by WHO SCC utilization (Varaganti et al. [12], Nababan et al. [11], and Tuyishim et al. [10]) (Table 5).

**9. Postpartum counseling.** There was 73 times more likelihood of counseling mothers on postpartum danger signs among professionals utilizing WHO SCC (OR 73.9, 95% CI 37–142.31, four studies, 1876 participants) (Low quality of evidence). Random effect meta-analysis was used for this outcome because of significant heterogeneity ($I^2 = 65\%$). The substantial heterogeneity indicates that treatment effects vary between studies, so we investigated the factors affecting treatment effects by a subgroup analysis of study design. Fixed effect meta-analysis of pre-and-post-intervention studies showed 132 times more postpartum danger sign counseling among professionals utilizing WHO SCC (OR 132.51, 95% CI 49.27–356.36, $I^2$ 0%, three studies, 1396 participants) (Fig 8).

**10. Counseling on family planning.** Five studies reported on counseling on family planning (Kumar et al. [7], Nababan et al. [11], Tuyishim et al. [10], Spector et al. [3], and

**Table 5. Newborn feeding assessment upon discharge with or without WHO SCC.**

| Study | Newborn feeding assessment upon discharge without WHO SCC (Number and %) | Newborn feeding assessment with WHO SCC (Number and %) | P-value |
|---|---|---|---|
| 1. Kumar et al. [7] | 31/240(13%) | 194/240(81%) | <0.001 |
| 2. Nababan et al. [11] | 152/153(99%) | 157/157(100%) | 0.318 |
| 3. Semrau et al. [21] | 2/1041(0.2%) | 225/1019(22.5%) | <0.001 |
| 4. Tuyishim et al. [10] | 89/101(88.1%) | 84/98(84.8%) | 0.540 |
| 5. Varaganti et al. [12] | 451/635(71.1%) | 458/620(73.8%) | 0.259 |
| 6. Spector et al. [3] | 211/338(62.5%) | 448/488(91.8%) | <0.001 |

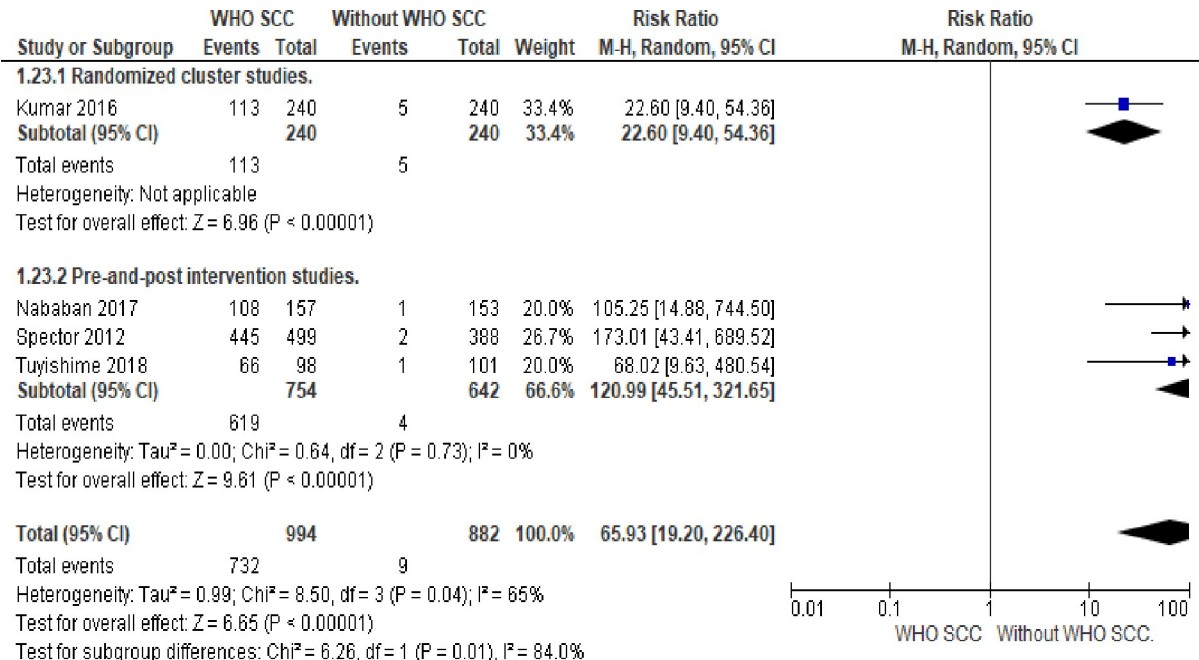

**Fig 8. Forest plot of comparison: 1 WHO SCC use and none use., outcome: 1.5 postpartum counseling.**

Varaganti et al. [12]). We didn't perform a meta-analysis because of significant heterogeneity that did not resolve by planned sensitivity analysis. Two studies uniformly reported significant improvement in counseling for family planning with WHO SCC utilization (Kumar et al. [7] and Spector et al. [3]). In contrast, two studies reported that there is no difference in counseling for postpartum family planning with or without WHO SCC (Varaganti et al. [12] and Tuyishim et al. [10]). Nababan et al. [11] reported that there is a statistically non-significant improvement in counseling for family planning with WHO SCC utilization (Table 6).

**11. Stillbirth.** Utilization of WHO SCC by health professionals reduces fresh stillbirth by 8% compared to none use of WHO SCC (OR 0.92, 95% CI 0.87–0.96, $I^2 = 0\%$, five studies, 299,952 participants, moderate quality of evidence) (Fig 9).

**12. Early neonatal death.** There is no statistically significant difference in early neonatal death with or without WHO SCC utilization (OR 1.07, 95% 0.01–1.13, $I^2 = 50\%$ five studies, 293,467 participants, very low quality of evidence). Random effect meta-analysis was utilized for this outcome because of heterogeneity ($I^2 = 50\%$) (Fig 10).

**Table 6. Counselling on family planning with or without WHO SCC utilization.**

| Study | Counseling on family planning without WHO SCC (Number and %) | Counseling on family planning with WHO SCC (Number and %) | P-value |
|---|---|---|---|
| 1. Kumar et al. [7] | 12/240(5%) | 103/240(43%) | <0.001 |
| 2. Nababan et al. [11] | 152/153(99%) | 157/157(100%) | 0.318 |
| 3. Spector et al. [3] | 1/338(0.3%) | 466/489(95%) | <0.001 |
| 4. Tuyishim et al. [10] | 94/100(93.1%) | 88/98(88.9%) | 0.333 |
| 5. Varaganti et al. [12] | 635/635(100%) | 620/620(100%) | - |

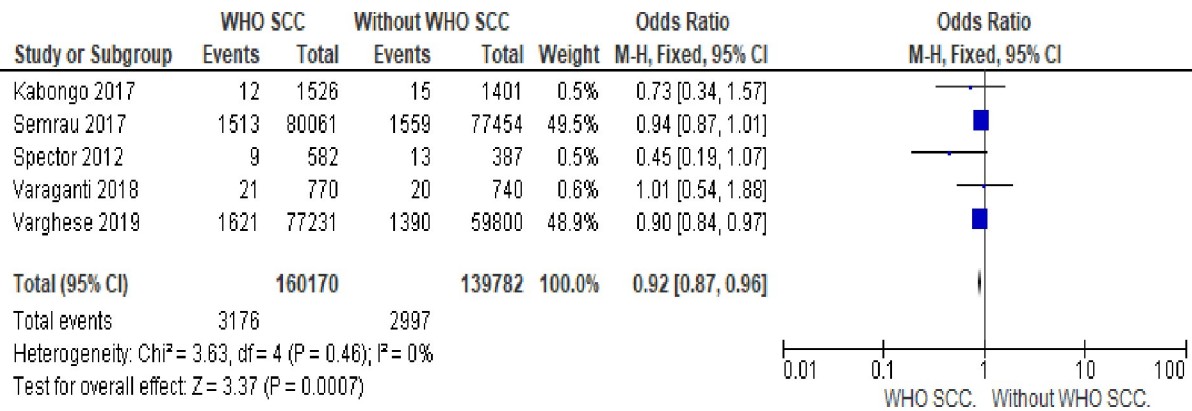

**Fig 9. Forest plot of comparison: 1 WHO SCC use and none use., outcome: 1.6 stillbirth.**

**13. Maternal death.** There is no statistically significant difference in maternal death with or without WHO SCC utilization (OR 1.06,95% 0.77–1.57, $I^2$ = 0% three studies, 159,934 participants, low quality of evidence) (Fig 11).

**14. Maternal morbidity.** One study (Semrau et.al [21]) reported that WHO SCC utilization has no statistical significant impact on maternal seizure (OR 0.93,95% 0.66–1.30), PPH (OR 0.94,95% 0.91–0.98), maternal sepsis (OR 1.02,95% 0.98–1.07), peri partum hysterectomy (OR 1.02,0.54–1.95) and blood transfusion (OR 0.99,0.89–1.11).

Methodological quality was assessed for seven outcomes using the GRADE approach (Shown in Table 7 below). The outcomes preeclampsia management, maternal infection management, partograph use, and stillbirth were assigned moderate-quality evidence scores. The outcomes postpartum danger sign counseling and maternal death were assigned low-quality evidence scores, where-as early neonatal death outcomes were assigned very low-quality evidence scores. The WHO SCC utilization was effective in improving the quality of essential childbirth practices like preeclampsia management, maternal infection management, post-partum danger sign counseling, partograph use, and breastfeeding practice within one hour of delivery.

## Discussion

This systematic review attempted to locate available evidence on the impact of WHO SCC utilization on essential childbirth practices and maternal and perinatal outcomes. Studies

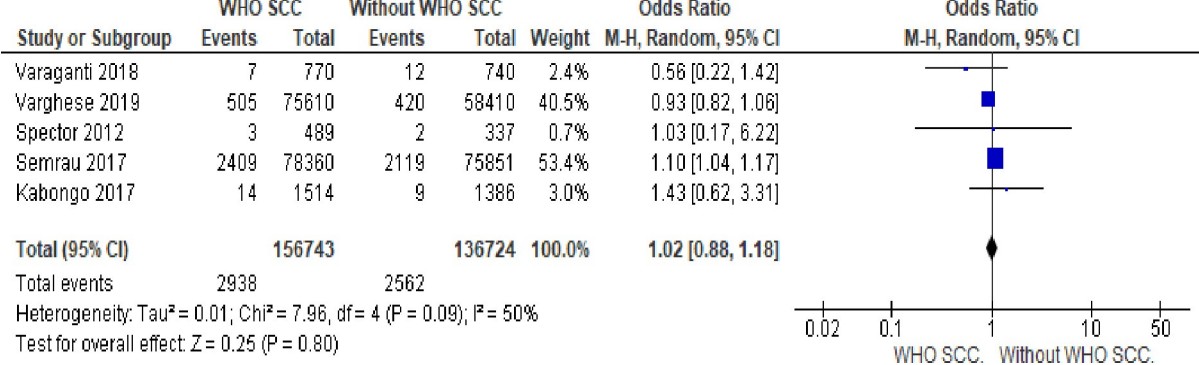

**Fig 10. Forest plot of comparison: 1 WHO SCC use and none use., outcome: 1.7 early neonatal death.**

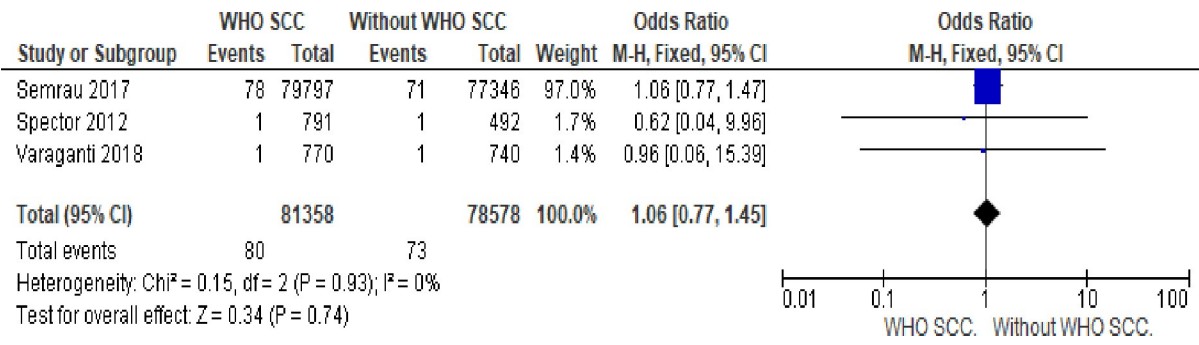

**Fig 11. Forest plot of comparison: 1 WHO SCC use and none use. Outcome: 1.7 maternal death.**

**Table 7. Summary of Findings(SOF).**

| Outcomes | Anticipated absolute effects* (95% CI) | | Relative effect (95% CI) | of participants (studies) | The certainty of the evidence (GRADE) |
|---|---|---|---|---|---|
| **WHO SCC compared to Usual care without WHO SCC for laboring mothers and newborn evaluation and management** | | | | | |
| **Patient or population: health professionals, laboring mothers, and newborns.** | | | | | |
| **Setting: sub-district, district, and tertiary health care.** | | | | | |
| **Intervention: WHO SCC** | | | | | |
| **Comparison: Usual care without WHO SCC** | | | | | |
| | The risk with Usual care without WHO SCC | The risk with WHO SCC | | | |
| **Preeclampsia management.** | 195 per 1,000 | 631 per 1,000 (362 to 838) | OR 7.05 (2.34 to 21.29) | 5667 (7 RCTs) | MODERATE [a] |
| **Maternal infection management.** | 110 per 1,000 | 805 per 1,000 (253 to 1,000) | RR 7.29 (2.29 to 23.27) | 5667 (7 RCTs) | MODERATE [a] |
| **Partograph.** | 190 per 1,000 | 472 per 1,000 (287 to 664) | OR 3.81 (1.72 to 8.43) | 5357 (6 RCTs) | MODERATE [a] |
| **Still birth.** | 21 per 1,000 | 20 per 1,000 (19 to 21) | OR 0.92 (0.87 to 0.96) | 299952 (5 RCTs) | MODERATE [b] |
| **Early neonatal death.** | 19 per 1,000 | 20 per 1,000 (19 to 21) | OR 1.07 (1.01 to 1.13) | 293467 (5 RCTs) | VERY LOW [b, c, d] |
| **Maternal death.** | 1 per 1,000 | 1 per 1,000 (1 to 1) | OR 1.06 (0.77 to 1.45) | 159936 (3 RCTs) | LOW [d, e] |
| **Postpartum counselling.** | 5 per 1,000 | 601 per 1,000 (223 to 1,000) | RR 132.51 (49.27 to 356.36) | 1876 (4 observational studies) | LOW [f, g] |

*The risk in the intervention group (and its 95% confidence interval) is based on the assumed risk in the comparison group and the relative effect of the intervention (and its 95% CI).

CI: Confidence interval; OR: Odds ratio; RR: Risk ratio.

GRADE Working Group grades of evidence.

High certainty: We are very confident that the actual effect lies close to that of the estimate of the impact.

Moderate certainty: We are moderately confident in the effect estimate: The real impact is likely to be close to the estimate of the effect, but there is a possibility that it is substantially different.

Low certainty: Our confidence in the effect estimate is limited: The actual impact may be significantly different from the estimate of the effect.

Very low certainty: We have very little confidence in the effect estimate: The exact result is likely to be substantially different from the estimate of effect.

**Explanations**

a. Five studies are pre-and -post-intervention studies and two cluster -randomized trials were included. Downgraded one level for risk of bias of included studies.

b. Two clusters -randomized, three pre-and-post intervention studies were included. Downgraded one level for risk of bias of included studies.

c. Lowered one level for inconsistent outcomes across studies.

d. Wide and statistically non-significant confidence interval.

e. One cluster-randomized trial and two pre-and -post-intervention studies were included. Downgraded one level for risk of bias of included studies.

f. One cluster-randomized and three pre-and -post-intervention studies were involved. Lowered one level for risk of included studies.

g. Wider confidence interval.

included in the review were three cluster randomized trials and six pre-and-post intervention studies. The pre- and -post-intervention studies did not undergo proper random allocation and allocation concealment and were judged to be at high risk of bias because of poor design. The effect estimates were also found to be high in most outcomes which might be reflecting underlying heterogeneity in set up and study design. However, the cluster-randomized studies and pre and post-intervention studies' results were consistent.

Moderate quality of evidence indicates that there was seven times more likelihood of evaluating labouring mothers for preeclampsia and administration of MgSo4 and antihypertensive drugs, 17 times more likely to evaluate and manage maternal infection, five times more likely to use partograph and 73 times more likelihood of counseling mothers on postpartum danger signs among professionals utilizing WHO SCC. Moderate quality of evidence also indicated that utilization of WHO SCC by health professionals reduces fresh stillbirth by 8% compared to none use of WHO SCC.

Low quality of evidence indicates that the utilization of WHO SCC has no impact on maternal death. Further studies are needed as only three primary studies were combined which might not reveal small changes since maternal death is a rare event. Very low quality of evidence indicates that the utilization of WHO SCC has no impact on early neonatal death. However, the primary studies included are of poor quality and moderately heterogeneous with $I^2 = 50\%$ mandating further well-designed studies.

Given the significant heterogeneity, combining of the outcomes active management of the third stage of labour, newborn assessment for sepsis, newborn feeding assessment upon discharge, maternal postpartum bleeding assessment, and postnatal family planning counseling was not possible. So, further well-designed studies are needed to generate evidence on those essential childbirth practices. Four studies uniformly reported significant improvement in AMTSL with WHO SCC utilization (Kumar et al. [7], Spector et al. [3], Varaganti et al. [12] and Semrau et al. [21]). In contrast, two studies reported statistically non-significant improvement (Nababan et al. [11] and Tuyishim et al. [10]). Two studies uniformly reported significant improvement in postpartum bleeding assessment with WHO SCC utilization (Kumar et al. [7] and Spector et al. [3]). In contrast, two studies reported statistically non-significant improvement (Varaganti et al. [12] and Nababan et al. [11]).

Two studies uniformly reported significant improvement in newborn assessment for sepsis with WHO SCC utilization (Kumar et al. [7] and Spector et al. [3]). In contrast, two studies reported statistically non-significant improvement (Varaganti et al. [12] and Tuyishim et al. [10]). Nababan et al. [11] reported that there is no difference in newborn assessment for sepsis with or without WHO SCC utilization (0.65% versus 0.64% with a p-value of 0.985). Three studies uniformly reported significant improvement in newborn feeding assessment upon discharge with WHO SCC utilization (Kumar et al. [7], Spector et al. [3], and Semrau et al. [21]). In contrast, three studies reported statistically non-significant improvement in newborn feeding assessment upon discharge by WHO SCC utilization (Varaganti et al. [12], Nababan et al. [11], and Tuyishim et al. [10]).

Two studies uniformly reported significant improvement in counseling for family planning with WHO SCC utilization (Kumar et al. [7] and Spector et al. [3]). In contrast, two studies reported that there is no difference in counseling for postpartum family planning with or without WHO SCC (Varaganti et al. [12] and Tuyishim et al. [10]). Nababan et al. [11] reported that there is a statistically non-significant improvement in counseling for family planning with WHO SCC utilization.

Only one randomized cluster study (Semrau et al. [21]) reported that WHO SCC utilization has no statistically significant impact on maternal seizure, PPH, maternal sepsis, peripartum

hysterectomy, and blood transfusion. This mandates further study to provide evidence on the impact of WHO SCC on maternal morbidity reduction.

# Conclusions

## Implications for practice

WHO SCC was effective in improving some of the essential child health practices and reducing stillbirth. Moderate quality of evidence indicates that WHO SCC is effective in) improving pre-eclampsia management b) improving maternal infection management c) improving parto-graph utilization d) reducing stillbirth. Low quality of evidence indicates that WHO SCC is effective in enhancing postpartum danger sign counseling. Low and very low quality of evidence suggests that WHO SCC has no impact on maternal and early neonatal death, respectively.

## Recommendations for research

The evidence regarding the effect of utilizing WHO SCC on active management of the third stage of labour, newborn assessment for sepsis, newborn feeding assessment before discharge, and postpartum family planning counseling is limited, heterogeneous, and of poor quality. Hence, further well-designed studies are needed to provide evidence on WHO SCC's impact on the above essential childbirth practices, and perinatal mortality, maternal death, and mater-nal morbidity.

# Supporting information

**S1 Document. Studies excluded and reasons for their exclusion.** It indicates excluded studies and reasons for their exclusion.
(DOCX)

**S1 Table. Search strategy for the MEDLINE database.** It indicates a detailed search strategy for PubMed.
(DOCX)

**S1 Checklist. PRISMA checklist.** It describes the review against the checklist for the PRISMA reporting guideline.
(DOCX)

# Acknowledgments

We are thankful to Saint Paul's Hospital Millennium Medical College center of excellency for reproductive health (SPHMMC COE) for coordinating training on systematic review.

# Author Contributions

**Conceptualization:** Lemi Belay Tolu, Garumma Tolu Feyissa.

**Data curation:** Lemi Belay Tolu, Garumma Tolu Feyissa.

**Formal analysis:** Lemi Belay Tolu, Garumma Tolu Feyissa.

**Funding acquisition:** Lemi Belay Tolu.

**Investigation:** Lemi Belay Tolu, Garumma Tolu Feyissa.

**Methodology:** Lemi Belay Tolu, Garumma Tolu Feyissa.

**Project administration:** Lemi Belay Tolu.

**Resources:** Lemi Belay Tolu, Garumma Tolu Feyissa.

**Software:** Lemi Belay Tolu, Garumma Tolu Feyissa.

**Supervision:** Lemi Belay Tolu, Wondimu Gudu Jeldu, Garumma Tolu Feyissa.

**Validation:** Lemi Belay Tolu, Wondimu Gudu Jeldu, Garumma Tolu Feyissa.

**Visualization:** Lemi Belay Tolu, Garumma Tolu Feyissa.

**Writing – original draft:** Lemi Belay Tolu, Garumma Tolu Feyissa.

**Writing – review & editing:** Lemi Belay Tolu, Garumma Tolu Feyissa.

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
