## [Decision Letter · Decision Letter 0]

8 Apr 2020

PONE-D-20-00206

Effectiveness of utilizing the WHO Safe Childbirth Checklist on improving essential childbirth practices and maternal and perinatal outcome: A systematic review and meta-analysis.

PLOS ONE

Dear Dr. Belay Tolu,

Thank you for submitting your manuscript to PLOS ONE. After careful consideration, we feel that it has merit but does not fully meet PLOS ONE’s publication criteria as it currently stands. Therefore, we invite you to submit a revised version of the manuscript that addresses the points raised during the review process.

Please address in the revised version of your manuscript all comments made by the reviewers, with a special attention to those about the study methods and statistical analysis.

Also, English language should be checked by a English native language author or external editing service.

Please refer to the instructions to authors of systematic reviews and meta-analyses on the Journal website, and the included PRISMA list. 

We would perfectly understand, would you consider that the changes required are too numerous or important to allow you to revise  the manuscript. Would you choose to revise, however, we would appreciate receiving your revised manuscript by May 23 2020 11:59PM. To enhance the reproducibility of your results, we recommend that if applicable you deposit your laboratory protocols in protocols.io, where a protocol can be assigned its own identifier (DOI) such that it can be cited independently in the future. For instructions see: http://journals.plos.org/plosone/s/submission-guidelines#loc-laboratory-protocols

We look forward to receiving your revised manuscript.

Kind regards,

Umberto Simeoni

Academic Editor

PLOS ONE

Journal Requirements:

2)  We noticed you have some minor occurrence(s) of overlapping text with the following previous publication(s), which needs to be addressed:

https://doi.org/10.2147/JMDH.S228951

https://doi.org/10.1371/journal.pone.0211298

https://doi.org/10.1002/14651858.CD007892.pub5

In your revision ensure you cite all your sources (including your own works), and quote or rephrase any duplicated text outside the Methods section. Further consideration is dependent on these concerns being addressed.

3) We note that you have indicated that data from this study are available upon request. PLOS only allows data to be available upon request if there are legal or ethical restrictions on sharing data publicly. For information on unacceptable data access restrictions, please see http://journals.plos.org/plosone/s/data-availability#loc-unacceptable-data-access-restrictions.

4) Please ensure that you refer to Figure 3 in your text as, if accepted, production will need this reference to link the reader to the figure.

Reviewers' comments:

Reviewer's Responses to Questions

**Comments to the Author**

1. Is the manuscript technically sound, and do the data support the conclusions?

Reviewer #1: Yes

Reviewer #2: Yes

Reviewer #3: Yes

2. Has the statistical analysis been performed appropriately and rigorously? 

Reviewer #1: Yes

Reviewer #2: I Don't Know

Reviewer #3: Yes

3. Have the authors made all data underlying the findings in their manuscript fully available?

Reviewer #1: Yes

Reviewer #2: Yes

Reviewer #3: Yes

4. Is the manuscript presented in an intelligible fashion and written in standard English?

Reviewer #1: Yes

Reviewer #2: Yes

Reviewer #3: Yes

5. Review Comments to the Author

Reviewer #1: I would like to mention the following comments:

1- Abstract is too long. The URLs are better not to present in abstract.

2- There is a confusion across the whole article. It is not known if it is a systematic review or prospective interventional study. It seems that the authors combined these two types of studies.

3- In the whole article, methods and inclusion/exclusion criteria of these two types of studies are confusing.

4- The version of EndNote is missing.

5- Figure 1: The details of databases (241) and other sources (217) are missing. In addition, the name of figure is not correct.

6- Table 1: Quality check of studies can be presented at a new column (last).

7- Only the quality of 8 studies has been discussed.

8- The studies with high percentage of Bias need more explanation for the solution.

9- Only two groups of "WHO SCC use and None use" have been presented in this article. How about the other methods?

10- Table 7: It is not known that the data in this table belong to systematic review or prospective interventional study.

Good Luck

Reviewer #2: This study presents a systematic review of the effectiveness of the using the WHO safe childbirth checklist (SCC) to improve child, maternal and perinatal outcomes. This study addresses an important research topic that would be of high interest. It would benefit from a more concise description of the research question and more critical discussion of the strengths and limitations of the study.

• Abstract – in the results are the measures of effect reported pooled estimates? If so, the meta-analysis process and pooling needs to be described.

• Abstract – the study outcomes need to be described in the research question

• Abstract – reference to the SOF table is not helpful in the abstract and it would be better if the results of the SOF could be summarized.

• Intro – the methods described at the bottom of page 4 and start of page 5 should be moved to the methods (not intro).

• Was the systematic review registered or is there a published protocol?

• The research question should clearly define the eligible study outcomes

• The two research questions are overlapping and it is unclear what the distinctions are? Is it the effectiveness vs efficacy or the different wording to describe the outcomes? The research question is more clearly described in the original PROSPERO registration. If it did not change after registration, I recommend using the registered question here.

• Under participants it would be helpful to know if there were any age, language, or country restrictions.

• The comparator section suggests use of other checklists was excluded. Were there any studies that compared two checklists that were excluded?

• Were the outcomes defined a priori and searched for or was this the list of outcomes that emerged from the search?

• Figure 1 requires a more detailed title and the reasons for exclusions should be provided.

• Lines 231 -235 are unclear

• What are the numbers in Table 1, column 5?

• Line 263. What is quasi-randomized? Do you mean quasi-experimental and non-randomized?

• Many of the effect estimates in the meta-analysis are incredibly large. Could this be discussed, and some context given?

• Why were meta-analyses not conducted for some of the outcomes with significant heterogeneity? It would be helpful to see all outcomes as meta-analysis when data are available.

• The discussion would benefit from some more critical review of the strengths and limitations of this study.

Reviewer #3: Abstract:it is not clear if authors included RCTs only or not

If not, as in methods, it is not clear if they pooled results from observational studies evalauted with multivariate analysis or not

methods (page 14): what does quasi xperimental studies mean?

methods: Sensitivity analyses were conducted by excluding studies with very large or very low effect estimates. Maybe authors should perform isntrad leave out analysis?

methods/results; in all analyss, despite inconsisency of 50% or more, authors used fixed effect. This should be changed into random.

results: authors should say that in all analysis rct and observational studies were consistent (it is a point of strenght)

6. PLOS authors have the option to publish the peer review history of their article (what does this mean?). If published, this will include your full peer review and any attached files.

Reviewer #1: No

Reviewer #2: No

Reviewer #3: Yes: Fabrizio D'Ascenzo

---

## [Author Response · Author response to Decision Letter 0]

7 May 2020

May 5, 2020

To: PLOS ONE Editor in chief.

Dear Editor in chief.

We would like to thank the reviewers for their thoughtful review of the manuscript. They raise important issues and their inputs are very helpful for improving the manuscript. We agree with almost all their comments and we have revised our manuscript accordingly. We respond below in detail to each of the reviewer’s comments. We hope that you find our responses satisfactory and that the manuscript is now acceptable for publication 

Looking forward hearing from you soon

Sincerely,

Lemi B Tolu (MD, Assistant prof of obstetrics and gynecology).

Saint Paul’s Millennium Medical College(SPHMMC)

Department of Obstetrics and Gynecology

Addis Ababa, Ethiopia.

Email: lemi.belay@gmail.com

Dear editor and reviewer 

Thanks for thoughtful review of the manuscript. Below is point by point response to raised concerns and how we changed the manuscript according to the comments.

Editor comments:

1. Please ensure that your manuscript meets PLOS ONE's style requirements, including those for file naming, English editing

Authors: Dear Editor thanks for the comment, the manuscript was revised accordingly, the grammar was editing further by two persons (almost rewritten).

2. We noticed you have some minor occurrence(s) of overlapping text with the following previous publication: https://doi.org/10.2147/JMDH.S228951, https://doi.org/10.1371/journal.pone.0211298,https://doi.org/10.1002/14651858.CD007892.pub5.

Authors: Dear Editor thank you very much those areas are rewritten, and reference was provided.

3. We note that you have indicated that data from this study are available upon request. PLOS only allows data to be available upon request.

Authors: Dear Editor the comment is well noted, all the data used in the analysis of the manuscript were included in manuscript and we revised accordingly.

Reviewer #1: 

1- Abstract is too long. The URLs are better not to present in abstract.

Authors: Dear reviewer this comment is well noted, and we revised abstract accordingly and removed URL.

2- There is a confusion across the whole article. It is not known if it is a systematic review or prospective interventional study. It seems that the authors combined these two types of studies.

3- In the whole article, methods and inclusion/exclusion criteria of these two types of studies are confusing.

Authors: Dear reviewer thank you very much for concerns on number 2 and 3 above, the comment made us to reevaluate our work and we have done a lot of changes throughout the manuscript especially on inclusion criteria’s and English editing’s. Regarding tables it’s our plan from the beginning to describe heterogeneous data’s by tables and narratively.

4- The version of EndNote is missing.

Authors: Dear reviewer the mistake well noted, and correction made (Line 74 and 76)

5- Figure 1: The details of databases (241) and other sources (217) are missing. In addition, the name of figure is not correct.

Authors: Dear reviewer thanks for the comment, we have revised the title of the figure, in this standard Prisma flow diagram we just indicated the source of our studies, the details of the source were included under search strategy.

6- Table 1: Quality check of studies can be presented at a new column (last).

Authors: Dear reviewer, comment is well noted, and we tried to do (add last column) but it goes beyond page and violates the journal requirement. We decided to narrate under methodological quality, thank you very much for understanding us.

7- Only the quality of 8 studies has been discussed.

Authors: Dear reviewer noted and corrected (Page 15, line 274-276).

8- The studies with high percentage of Bias need more explanation for the solution.

Authors: Dear reviewer the methodology of all studies were explained under six domains in detail (page 15-17, lines 274-318).

9- Only two groups of "WHO SCC use and None use" have been presented in this article. How about the other methods?

Authors: Dear reviewer we found WHO SCC is the one widely used, especially in low and middle-income countries and we didn’t come across studies that report on other checklist fulfilling our inclusion criteria (Page 13, line 256).

10- Table 7: It is not known that the data in this table belong to systematic review or prospective interventional study.

Authors: Dear Reviewer Table 7 is summary of finding table, generated by online Gradepro (we exported data directly from Revman to Gradepro). We have included health professionals under participants in the table considering the comment. Thank you very much for understanding.

Reviewer #2

This study presents a systematic review of the effectiveness of the using the WHO safe childbirth checklist (SCC) to improve child, maternal and perinatal outcomes. This study addresses an important research topic that would be of high interest. It would benefit from a more concise description of the research question and more critical discussion of the strengths and limitations of the study.

Authors: thank you very much dear reviewer, we have addressed your comments accordingly to make the manuscript much better.

1. Abstract – in the results are the measures of effect reported pooled estimates? If so, the meta-analysis process and pooling needs to be described.

2- Abstract – the study outcomes need to be described in the research question

3- Abstract – reference to the SOF table is not helpful in the abstract and it would be better if the results of the SOF could be summarized.

Authors: Dear reviewer the above three comments are on abstract. The concerns were addressed, SOF table reference removed, the study outcomes included in research question and method of meta-analysis was described (Page 1-2, lines 22-24,31-32,25-46)

4- Intro – the methods described at the bottom of page 4 and start of page 5 should be moved to the methods (not intro).

Authors: Dear reviewer comment accepted and modified accordingly (Page 5and 6, lines 109-117) 

5- Was the systematic review registered or is there a published protocol?

Authors: yes, registered on Prospero ( available at : https://www.crd.york.ac.uk/PROSPERO/display_record.php?RecordID=137092)

6- The research question should clearly define the eligible study outcomes

7- The two research questions are overlapping, and it is unclear what the distinctions are? Is it the effectiveness vs efficacy or the different wording to describe the outcomes? The research question is more clearly described in the original PROSPERO registration. If it did not change after registration, I recommend using the registered question here.

Authors: Dear reviewer the concerns on research questions (Concern 6 and7) were valid and good input for us, we revised accordingly (Page 6, lines 124 and 127)

8- Under participants it would be helpful to know if there were any age, language, or country restrictions.

Authors: Dear reviewer we included studies published in English (doesn’t matter whatever language they speak if the study was published in English).

9- The comparator section suggests use of other checklists was excluded. Were there any studies that compared two checklists that were excluded?

Authors: Dear reviewer we didn’t come across checklist other than WHO SCC, the studies excluded were among WHO SCC who has no outcome of our interest and study design has no comparator.

10- Were the outcomes defined a priori and searched for or was this the list of outcomes that emerged from the search?

Authors: Dear reviewer all are predefined in the protocol, except some edits and minor modifications as we come across different evidences and literatures.

11- Figure 1 requires a more detailed title and the reasons for exclusions should be provided.

Authors: Thanks, dear Reviewer, title corrected and reason for exclusion added (Page 12, line 241-244). 

12- Lines 231 -235 are unclear

Authors: Dear reviewer the certainty of level of quality of evidence was assessed using Gradepro. We did grading for seven outcomes. We have made change to the sentences to make clearer (Page 11, lines 231-235)

13- What are the numbers in Table 1, column 5?

Authors: Dear Reviewer, its number of participants in the intervention and comparator group, we made the statement clearer.

14- Line 263. What is quasi-randomized? Do you mean quasi-experimental and non-randomized?

Authors: Dear reviewer yes, it’s to mean non-randomized, its corrected as cluster randomized.

15- Many of the effect estimates in the meta-analysis are incredibly large. Could this be discussed, and some context given?

Authors: Dear reviewer we tried to address in the discussion part as might be because of heterogeneity (Line 511-513) 

16- Why were meta-analyses not conducted for some of the outcomes with significant heterogeneity? It would be helpful to see all outcomes as meta-analysis when data are available.

Authors: Dear reviewer we planned to pull together those without significant heterogeneity (I2 < 50%), as pulling together significant heterogenous studies for getting statistics might be introducing more bias to the evidence synthesis. That’s why we didn’t do pull them together, thanks for understanding.

17- The discussion would benefit from some more critical review of the strengths and limitations of this study.

Authors: Dear reviewer thank you very much the, we added sentence to make it clearer to first paragraph of discussion which has strength and limitations (Page 30, lines 505-508 and lines 524-528). 

Reviewer#3

2. Abstract: it is not clear if authors included RCTs only or not. If not, as in methods, it is not clear if they pooled results from observational studies evaluated with multivariate analysis or not

Authors: Dear reviewer we have included types of studies. They were cluster randomized and pre and post intervention studies. 

3. Methods (page 14): what does quasi experimental studies mean?

Authors: Dear reviewer It was to describe cluster randomized studies and corrected like that (Line 281).

4. methods: Sensitivity analyses were conducted by excluding studies with very large or very low effect estimates. Maybe authors should perform instead leave out analysis?

Authors: Dear reviewer what we did is like leave out analysis, we excluded and see its impact on the heterogeneity (corrected line 227).

5. methods/results; in all analyses, despite inconsistency of 50% or more, authors used fixed effect. This should be changed into random.

Authors: Dear reviewer thank you very much, we have noted the mistake and corrected (Figure 8 on page 26 and Figure 10 on page 28)

6. results: authors should say that in all analysis rct and observational studies were consistent (it is a point of strenght)

Authors: Dear Reviewer thank you very much, included in the discussion part (page 30, Lines 509)

---

## [Decision Letter · Decision Letter 1]

26 May 2020

Effectiveness of utilizing the WHO Safe Childbirth Checklist on improving essential childbirth practices and maternal and perinatal outcome: A systematic review and meta-analysis.

PONE-D-20-00206R1

Dear Dr. Tolu,

We are pleased to inform you that your manuscript has been judged scientifically suitable for publication and will be formally accepted for publication once it complies with all outstanding technical requirements.

With kind regards,

Umberto Simeoni

Academic Editor

PLOS ONE

Additional Editor Comments (optional):

Reviewers' comments:

Reviewer's Responses to Questions

**Comments to the Author**

1. If the authors have adequately addressed your comments raised in a previous round of review and you feel that this manuscript is now acceptable for publication, you may indicate that here to bypass the “Comments to the Author” section, enter your conflict of interest statement in the “Confidential to Editor” section, and submit your "Accept" recommendation.

Reviewer #1: All comments have been addressed

Reviewer #3: All comments have been addressed

2. Is the manuscript technically sound, and do the data support the conclusions?

Reviewer #1: Yes

Reviewer #3: (No Response)

3. Has the statistical analysis been performed appropriately and rigorously? 

Reviewer #1: Yes

Reviewer #3: (No Response)

4. Have the authors made all data underlying the findings in their manuscript fully available?

Reviewer #1: Yes

Reviewer #3: (No Response)

5. Is the manuscript presented in an intelligible fashion and written in standard English?

Reviewer #1: Yes

Reviewer #3: (No Response)

6. Review Comments to the Author

Reviewer #1: The authors have considered all my comments. They answered all comments one by one. I agree to accept the article.

Reviewer #3: (No Response)

7. PLOS authors have the option to publish the peer review history of their article (what does this mean?). If published, this will include your full peer review and any attached files.

Reviewer #1: Yes: Masoud Amiri

Reviewer #3: Yes: Fabrizio D'Ascenzo

---

## [Editor Report · Acceptance letter]

28 May 2020

PONE-D-20-00206R1 

Effectiveness of utilizing the WHO Safe Childbirth Checklist on improving essential childbirth practices and maternal and perinatal outcome: A systematic review and meta-analysis 

Dear Dr. Tolu:

I am pleased to inform you that your manuscript has been deemed suitable for publication in PLOS ONE. Congratulations! Your manuscript is now with our production department. 

With kind regards,

on behalf of

Dr. Umberto Simeoni 

Academic Editor

PLOS ONE